# Night Vision Anti-Halation Method Based on Infrared and Visible Video Fusion

**DOI:** 10.3390/s22197494

**Published:** 2022-10-02

**Authors:** Quanmin Guo, Hanlei Wang, Jianhua Yang

**Affiliations:** School of Electronic and Information Engineering, Xi’an Technological University, Xi’an 710021, China

**Keywords:** night vision anti-halation, video fusion, infrared image, visible image, frame selection strategy, adaptive motion compensation

## Abstract

In order to address the discontinuity caused by the direct application of the infrared and visible image fusion anti-halation method to a video, an efficient night vision anti-halation method based on video fusion is proposed. The designed frame selection based on inter-frame difference determines the optimal cosine angle threshold by analyzing the relation of cosine angle threshold with nonlinear correlation information entropy and de-frame rate. The proposed time-mark-based adaptive motion compensation constructs the same number of interpolation frames as the redundant frames by taking the retained frame number as a time stamp. At the same time, considering the motion vector of two adjacent retained frames as the benchmark, the adaptive weights are constructed according to the interframe differences between the interpolated frame and the last retained frame, then the motion vector of the interpolated frame is estimated. The experimental results show that the proposed frame selection strategy ensures the maximum safe frame removal under the premise of continuous video content at different vehicle speeds in various halation scenes. The frame numbers and playing duration of the fused video are consistent with that of the original video, and the content of the interpolated frame is highly synchronized with that of the corresponding original frames. The average FPS of video fusion in this work is about six times that in the frame-by-frame fusion, which effectively improves the anti-halation processing efficiency of video fusion.

## 1. Introduction

Due to the halation phenomenon caused by the abuse of high beam lights at night, drivers approaching from opposite directions are unable to see the road conditions clearly, leading to potential traffic safety hazards [1].

Passive anti-halation methods are simple and effective, such as placing shading boards and growing plants on the middle isolation belts of two-way lanes [2]. However, it is often difficult to use these methods on a larger scale due to the limitations caused by road planning and other factors. Therefore, active anti-halation methods have attracted extensive attention from the research community.

In comparison with other active anti-halation methods, such as placing polarization film on the front windshield [3], infrared night vision imaging system [4], and dual CCD image sensors to expand the dynamic range of acquisition [5], the infrared and visible image fusion methods [6] combine the advantages of infrared images without halation and visible images containing rich color details; the resulting fused image has very minute halation and a good visual effect. However, night vision halation images belong to typical backlighting images with low illumination and strong light sources. The high brightness of the halation area overwhelms the effective information, while the brightness of the non-halation area is too low to observe the information in dark areas. Therefore, the improved IHS-Curvelet fusion method [7] not only eliminates the halation, but also enhances details such as color, texture and others in dark area, resulting in high computational complexity, which is only suitable for processing static images. When applied to video, the fusion efficiency is low, thus resulting in the anti-halation video lag.

Frame extraction technology can reduce the redundant frames of video to a certain extent and improve the fusion efficiency. The frame selection method based on sparse representation [8] is simple and easy to operate. However, the accuracy of the model is low for video frames with nonlinear structure. The retained frame extracted based on clustering [9] has a small redundancy and a strong ability to reflect the original video, but the temporal sequence of each frame is not considered in the processing. The motion analysis method [10] takes into account the motion characteristics of objects, which has a strong universality. The extracted frames have a high expression for contents of original video.

In order to address the problem that the frame rate of the extracted video is inconsistent with that of original video, it is necessary to make interpolation compensation for the retained frames to improve the visual smoothness. The laconic smooth technique based on multi-frame transformation asynchronously (LSTMTA) [11] is simple to implement. However, the quality of interpolated frames depends on its adjacent frames, and the effect is unstable. The frame insertion method based on optical flow and frame-recurrent network (OFFRN) [12] has high accuracy in detecting and tracking the position of moving targets. However, it assumes that the adjacent frames have constant brightness and small movement [13,14], and is unsuitable for the night vision halation scene in this paper. The motion vector estimated by block matching search (MVEBMS) [15] has the advantages of simple implementation and high processing efficiency, but it has block artifacts [16]. The frame interpolation method based on deep learning [17] mines depth features to obtain better visual quality, but has a large time overhead. It is difficult to meet real-time requirements when applied to video fusion in a night halation scene.

In this work, an efficient anti-halation method suitable for video fusion is presented. Considering the characteristics of small differences in content and high redundancy between adjacent video frames, a frame selection based on inter-frame difference (FSIFD) is designed to minimize the number of fused frames under the premise of continuous video content for improving the fusion efficiency. In addition, a time-mark-based adaptive motion compensation (TMBAMC) is designed for restoring the length of the anti-halation video to be consistent with the original video, and ensuring the content synchronization of corresponding frames. The proposed method is applied to a vehicle head-up display (HUD), which can produce a video without halation, with clear details and rich colors, to assist drivers in driving safely under a special night-time halation scene. In addition, it can be applied to advanced driver assistance systems (ADAS) or autonomous vehicles (AV) to improve the vehicle’s environmental perception ability in special scenarios.

The remaining text of the article is arranged as follows. Section 2 presents anti-halation principles and methods for infrared and visible video fusion. Section 3 describes a step-by-step design of the frame selection fusion strategy. Section 4 shows the realization of frame interpolation. Section 5 gives the experiential results and discussion. Lastly, Section 6 represents the conclusion.

## 2. Principle and Method

The anti-halation method of infrared and visible image fusion makes use of the complementarity of different source images. The resulting fused image contains minimal halation and rich texture details, as shown in Figure 1. Based on the image fusion, an anti-halation method suitable for video fusion is designed to ensure the quality of the fused image and the continuity of video playing.

It is notable that there are only small differences in the content between adjacent frames of videos. This results in a lot of redundant operations in achieving the anti-halation image by frame-by-frame fusion, consequently leading to low efficiency. In order to address this issue, the FSIFD strategy is designed by analyzing the motion information of objects between adjacent frames. The strategy ensures that the redundant frames are discarded to the maximum extent in the original video under continuous video content, and only the retained frames are fused by the improved IHS-Curvelet algorithm to meet the high efficiency requirements.

In order to solve the problem of the shorter playing duration of the fused video caused by the reduced frames, this work proposes a TMBAMC algorithm to interpolate new frames. Considering the frame number as a time stamp, the interpolated frames with the same time and quantity as the discarded redundant frames are constructed, so that the number of frame and length of the fused video are restored to be consistent with that of the original video. In order to synchronize the content of the interpolated frames with that of the corresponding original frames, we take the motion vector of the two adjacent retained frames as the benchmark, construct the adaptive weights according to the inter-frame difference between the interpolated and the last retained frame, then estimate the motion vector of the interpolated frame.

The overall block diagram of the proposed night vision anti-halation method is presented in Figure 2.

## 3. Fusion Strategy of Frame Selection Based on Inter-Frame Difference

### 3.1. Quantization of Inter-Frame Difference

The difference between two adjacent frames is often measured based on Euclidean distance or cosine similarity. Compared to the Euclidean distance [18] that calculates the distance between two points in a multi-dimensional space, the cosine similarity [19] measures the inter-frame difference according to the cosine of the angle between two vectors. The cosine similarity is computationally efficient and perceives slight differences between video frames. Therefore, it is more suitable for applications requiring efficiency and strong correlation between video frames, and is adopted in this work.

The three-dimensional information of a color image is reduced to a one-dimensional array by gray partition. The gray levels in the range of [0,255] in each channel are divided into four intervals, including [0,63], [64,127], [128,191], and [192,255]. Then, the intensity *PV_i_* of the *i*th pixel in a frame is expressed as:(1)$$PV_{i}=\left\lfloor B_{i}/64\right\rfloor \times 4^{2}+\left\lfloor G_{i}/64\right\rfloor \times 4^{1}+\left\lfloor R_{i}/64\right\rfloor \times 4^{0}$$
where *PV_i_* ∈ [0, 63], *i* ∈ [0, L − 1], and L represents the total number of pixels in a frame. *R_i_*, *G_i_*, and *B_i_* represent the intensity of red, green, and blue channels at *i*th pixel, respectively. ⌊•⌋ denotes the floor operation.

The number *n_PVi_* of pixels corresponding to 64 intensity values in a frame is counted to form a feature vector as:(2)$$N=[n_{0}\;n_{1}\;\cdots \;n_{63}]$$

The difference between two arbitrary frames is expressed as the cosine angle *θ**_XY_*** of their vectors ***X*** and ***Y***:(3)$$\theta_{XY}=\arccos(\frac{X\cdot Y}{\left\|X\right\|\times \left\|Y\right\|})=\arccos(\frac{\sum_{j=0}^{63}x_{j}y_{j}}{\sqrt{\sum_{j=0}^{63}x_{j}^{2}}\times \sqrt{\sum_{j=0}^{63}y_{j}^{2}}})$$
where ***X*** = [*x*_0_ *x*_1_ … *x*_63_] and ***Y*** = [*y*_0_ *y*_1_ … *y*_63_]. The closer *θ**_XY_*** is to 0, the closer cos*θ**_XY_*** is to 1, and the higher the similarity of the two images, the smaller the difference.

### 3.2. Setting of Cosine Angle Threshold

The difference between frames is compared with the cosine angle threshold *τ* to discard the redundant frames in the video sequence. Therefore, the *τ* directly affects the number of frames removed from a video. If *τ* is too high, the video will become discontinuous due to too many frames removed, resulting in flickering and skipping. On the contrary, if *τ* is too small, there will still be redundant frames in the video sequence, leading to poor computational efficiency. Therefore, the threshold *τ* of cosine angle should meet the requirement of maximum of discarded frames on the premise of continuous video content.

We introduce an objective indicator of overall continuity of a video, namely nonlinear correlation information entropy (NCIE) [20,21], and define a de-frame rate (DFR). By analyzing the relations of *τ* with NCIE and DFR, the optimum threshold is determined to satisfy the maximum of discarded frames under the premise of continuous video content.

For a group of video sequences containing *K* frames, let the vectors of frames *m* and *w* be ***M*** = [*m*_0_ *m*_1_ … *m*_63_], ***W*** = [*w*_0_ *w*_1_ … *w*_63_]. Then, each data pair (*m_j_*, *w_j_*) is characterized in an *r* × *c* two-dimensional network, where *j* ∈ [0,63], *r* and *c* denote the number of rows and columns in the two-dimensional data respectively, where 1 ≤ *r* = *c* ≤ 8. The nonlinear correlation coefficient *NCC_mw_* between the *m*th frame and the *w*th frame is expressed as:(4)$$NCC_{mw}=NCC(M,W)=2+\sum_{r=1}^{8}\sum_{c=1}^{8}P_{rc}\log_{8}P_{rc}$$
where *P_rc_* denotes the joint probability distribution of ***M*** and ***W***, *P_rc_* = *Q_rc_*/64, and *Q_rc_* is the number of data pairs in the (*r*, *c*) th two-dimensional grid. Considering *NCC_mw_* as an element, a nonlinear correlation matrix *R* is formed as:(5)$$R={\left[NCC_{mw}\right]}_{1\leq m\leq K,1\leq w\leq K}$$

Now, the overall correlation degree NCIE of *K*-frame video sequence is expressed as:(6)$$\mathrm{NCIE}=1+\sum_{z=1}^{K}\frac{\lambda_{z}}{K}\log_{K}\frac{\lambda_{z}}{K}$$
where *λ_z_* denotes the *z*th eigenvalue of nonlinear correlation matrix *R*, *z* ∈ [1, *K*]. Please note that the larger the NCIE, the higher the content continuity of the video sequence.

The DFR is now mathematically expressed as:(7)$$\mathrm{DFR}=\frac{F_{R}}{K}\times 100\%$$
where *F_R_* denotes the number of redundant frames. The larger the DFR, the more frames are removed from the video sequence.

In order to make the threshold *τ* universal at different vehicle speeds in various scenes, a different *τ* is set to calculate the corresponding DFR and NCIE, respectively, for vehicle fast speed video (Fast video1) and vehicle slow speed video (Slow video1) containing suburban roads, as well as vehicle fast speed video (Fast video2) and vehicle slow speed video (Slow video2) comprising urban main roads; the results are shown in Figure 3.

It is evident from Figure 3a that with an increase in *τ*, the DFR also increases for all video sequences. However, please note that for the same threshold, the DFR corresponding to slow video is higher as compared to that of fast video.

It can be seen from Figure 3b that the fluctuation of NCIE is very small in *τ* ∈ [0,2], which is close to that at *τ* = 0, i.e., the case that no frame is removed; the NCIE drops sharply in *τ* ∈ [2,2.5]; the NCIE oscillates when *τ* > 2.5 but is generally much lower than that in *τ* ∈ [0,2.5]. It is notable that there is an inflection point in *τ* ∈ [1.5,2.5], which leads to the mutation of NCIE, thus weakening the overall correlation of video sequence and causing video discontinuity.

Therefore, the threshold *τ* is further determined in the interval [1.5,2.5] so that it has a certain margin from the inflection point of NCIE to ensure the continuity of video and a high frame removal rate. Figure 4 shows the NCIE in *τ* ∈ [1.5,2.5].

It is evident from Figure 4 that the NCIE mutates at *τ* = 2.1 and the video starts to become discontinuous. The NCIE is relatively stable in *τ* ∈ [1.5,2.1]. Therefore, the optimal cosine angle threshold *τ*_op_ is selected to be 1.8, i.e., located in the middle of the stationary region with a certain margin from the mutation point. This ensures the minimum of frames to be fused under the premise of a continuous video sequence.

### 3.3. Implementation of Frame Selection Fusion

For a group of video sequences containing *K* frames, starting from the first frame, the first frame is regarded as the reference frame *R* and is retained. The second and third frames are regarded as the current frame *C* and the next frame *F*, respectively. The inter-frame difference *θ**_RC_*** between *R* and *C* and the *θ**_RF_*** between *R* and *F* are computed and compared with the optimal threshold *τ*_op_ to determine if the *C* is retained or redundant.

If *θ**_RC_*** and *θ**_RF_*** are both smaller than *τ*_op_, *C* is regarded as a redundant frame and discarded. In this case, *R* stays the same in the next cycle. On the other hand, if *θ**_RC_*** is less than *τ*_op_ and *θ**_RF_*** is greater than *τ*_op_, *C* is regarded as the retained frame, and serves as the reference frame *R* during the next iteration. At the same time, *C* and *F* both move backwards by one frame, until *C* is the last frame in the sequence. Here, *C* is selected as the retained frame and the frame selection process is terminated.

Z*_R_*, Z*_C_*, and Z*_F_* represent the sequence number of *R*, *C*, and *F* in the original video, respectively, *count* denotes the retained frame counter, and the sequence number of the retained frame is stored in array *S*. The frame selection strategy is expressed as follows:(1)Initialization:*S*[0]←*Z_R_*←1;*Z_C_*←2;*Z_F_*←3;*count*←1;
(2)Iteratively select retained frames:while *Z_C_* < *K*{   if (*θ**_RC_** < τ*_op_)   {      if (*θ**_RF_*** > *τ*_op_)      {         *Z_R_*←*Z_C_*;            *S*[*count*]←*Z_C_*;            *count*←*count* + 1;      }      else      {         *Z_C_*←*Z_C_* + 1;         *Z_F_*←*Z_F_* + 1;      }   }}
(3)Keep the last frame and end the frame selection.*S*[*count* + 1]←*K*.

The above frame selection process ensures that the inter-frame difference between the retained frames is the highest, but not greater than the cosine angle threshold *τ*.

For the selected retained frame sequence, the improved IHS-curvelet algorithm [7] is used for performing anti-halation fusion. As the halation information is mainly distributed in the brightness component, the algorithm only performs single-channel fusion between the brightness component I of the visible image and the infrared image, thus reducing the computational complexity. The hue (H) and saturation (S) do not participate in the fusion, so as to avoid color distortions in the fused image.

The anisotropy of the support interval of Curvelet transform is utilized to achieve an efficient expression of two-dimensional information. The automatic adjustment of low-frequency coefficient weight is adopted to avoid the halation information from contributing to the reconstruction.

The brightness component I and the infrared image are decomposed by two-dimensional discrete Curvelet transform [22,23]. Their low-frequency coefficients and multiple high-frequency coefficients at different scales and directions are obtained as follows:(8)$$c^{D}(j,l,k)=\sum_{0\leq t_{1},t_{2}<n}f[t_{1},t_{2}]\sqrt{\varphi_{j,l,k}^{D}[t_{1},t_{2}]}$$
where *f* [*t*_1_, *t*_2_] is the input of the Cartesian coordinate system, $\varphi_{j,l,k}^{D}[t_{1},t_{2}]$ is the Curvelet function, where *D* represents discretization, *l* represents direction, *k* represents position, and *j* represents the scale of Curvelet decomposition.

The infrared low-frequency coefficient weights *α*(*k*_1_, *k*_2_) are mathematically expressed as:(9)$$a(k_{1},k_{2})=\frac{1}{2\pi }\arctan(l\cdot (c_{0}^{VI}(k_{1},k_{2})-m))+n$$
where $c_{0}^{VI}$ (*k*_1_, *k*_2_) is the low-frequency coefficient of visible image, *m* is the critical value at the junction of halation and non-halation in the low-frequency coefficient matrix, *n* is the weight of the infrared low-frequency coefficient at the critical value *m*, *l* is the critical rate of change, reflecting the intensity of change in *α*(*k*_1_, *k*_2_) at the junction of halation and non-halation. After multiple optimization and comparison, when *m* is 3, *n* is 0.75, and *l* is 2, the image fusion results reach the optimal level.

The low-frequency coefficient $c_{0}^{FU}$ (*k*_1_, *k*_2_) of the fused image is expressed as:(10)$$c_{0}^{FU}(k_{1},k_{2})=a(k_{1},k_{2})c_{0}^{IR}(k_{1},k_{2})+[1-a(k_{1},k_{2})]c_{0}^{VI}(k_{1},k_{2})$$
where $c_{0}^{IR}$ (*k*_1_, *k*_2_) is the infrared low-frequency coefficients.

The high-frequency coefficient $c_{j,l}^{FU}$ (*k*_1_, *k*_2_) of the fused image adopts the modulus maximization for containing more detailed information:(11)$$c_{j,l}^{FU}(k_{1},k_{2})=\max\left\{\left|c_{j,l}^{IR}(k_{1},k_{2})\right|,\left|c_{j,l}^{VI}(k_{1},k_{2})\right|\right\}$$
where $c_{j,l}^{IR}$ (*k*_1_, *k*_2_) and $c_{j,l}^{VI}$ (*k*_1_, *k*_2_) are the high-frequency coefficients of the infrared image and the brightness component I, respectively.

The $c_{0}^{FU}$, (*k*_1_, *k*_2_) and $c_{j,l}^{FU}$, (*k*_1_, *k*_2_) are reconstructed by discrete curvelet transform in the frequency domain to obtain the new brightness component I′. The discrete Curvelet transform in the frequency domain can be expressed as follows:(12)$$L(j,l,k)=\frac{1}{{\left(2\pi \right)}^{2}}\sum \hat{f}[\omega_{1},\omega_{2}]\bar{{\hat{\varphi }}_{j,l,k}[\omega_{1},\omega_{2}]}$$
where $\hat{f}$ [*ω*_1_, *ω*_2_] represents input in the frequency domain, and $\hat{\varphi }$*_j_*_,*l*,*k*_[*ω*_1_, *ω*_2_] is the Curvelet function in the frequency domain.

The IHS inverse transform is performed with the new brightness component I′, the original H and S. The resulting anti-halation fused image has very small halation and possesses rich details, such as edge contours and colors.

## 4. Time-Mark-Based Adaptive Motion Compensation Algorithm

In order to ensure that the frame number, frame rate, and playing duration of the fused video are consistent with that of the original video, this work considers the motion vector estimated by block matching search (MVEBMS) [15] as the benchmark, taking the sequence number of each retained frame as the time stamp to determine the interpolated frames, and constructs adaptive weights according to the difference between the interpolated and retained frames to estimate the motion vector of the interpolated frames.

Searching the block *g*(*x*,*y*) which has the minimum matching error with the current block *f*(*x*,*y*) in the reference frame range as the matching block, the relative displacement between *g*(*x*,*y*) and *f*(*x*,*y*) denotes the estimated motion vector ***MV***. The block matching search is shown in Figure 5. The minimum matching error *SAD*(*v_x_*,*v_y_*) is expressed as:(13)$$SAD(v_{x},v_{y})=\frac{1}{M\times N}\sum_{i=0}^{M-1}\sum_{j=0}^{N-1}|f(i,j)-g(i+v_{x},j+v_{y})|$$
where block size is *N* × *M* = 16 × 16 pixels and search window have size ±8 pixels. *i* and *j* are the horizontal and vertical coordinates of pixels, respectively. *v_x_* and *v_y_* are the horizontal and vertical components of ***MV***, respectively.

Let us assume that the front frame of two adjacent retained frames is FR and the back frame is BR. Then, the total number *T* of interpolated frames between them is:(14)$$T=Z_{\mathrm{FR}}-Z_{\mathrm{BR}}-1$$
where *Z*_FR_ and *Z*_BR_ are the sequence numbers of FR and BR, respectively. Considering the content difference *θ**_FB_*** between FR and BR as the reference, the adaptive weight *λ_i_* of the *i*th interpolation frame S*_i_* is expressed as:(15)$$\lambda_{i}=\frac{\theta_{FS_{i}}}{\theta_{FB}},\;0\leq \theta_{FS_{i}}<\theta_{FB}$$
where *θ**_FSi_*** is the content difference between FR and *S_i_*, 0 ≤ *i* ≤ *T*.

The estimated motion vector ***MV****_Si_* of the interpolated frame *S_i_* can be expressed as follows:(16)$$MV_{S_{i}}=\lambda_{i}MV$$

According to Equation (16), the pixel information of the interpolated frame is constructed. In order to satisfy the rate conversion and the visual effect of frame interpolation, the overlapping block motion compensation (OBMC) [24,25] is selected with low computational complexity and high frame insertion quality. The *i*th frame *F*(*x*,*y*,*i*) to be interpolated is determined:(17)$$F(x,y,i)=\sum_{j=1}^{M}W(n,m)*F_{j}(x+v_{x},y+v_{y},i-1)$$
where *x* and *y* are the horizontal and vertical coordinates of the frame to be interpolated, respectively. *v_x_* and *v_y_* are the horizontal and vertical components of ***MV***_S*i*_, respectively. *n* and *m* are the relative coordinates of horizontal and vertical positions in the window function, respectively. *j* is the number of overlapping blocks ranging in [1, M]. *W* (*n*, *m*) is the weight of pixel (*n*,*m*) in the window, expressed as:(18)W(n,m)=W(n)∗W(m)
where,
(19)$$W(n)=\frac{1}{2}(1-\cos(\pi \times \frac{n+1/2}{16}))\;n=0,1,2,3\ldots$$
(20)$$W(m)=\frac{1}{2}(1-\cos(\pi \text{×}\frac{m+1/2}{16}))m=0,1,2,3\ldots$$

## 5. Results and Discussion

In order to verify the effectiveness and universality of the anti-halation method based on infrared and visible video fusion, we collected infrared and visible videos on two typical roads covering common traffic at night, namely suburban roads and urban main roads. The videos include fast-speed vehicle videos and slow-speed vehicle videos in each scene.

In the suburban road scene, there are almost no other light sources except vehicle beams, and the overall illumination is very low. In the urban main road scenario, the scattered light from streetlamps and surrounding buildings is weak, and the vehicles are using low beam lights. The halation areas of the videos increase in size and then shrink as the vehicles come closer. We have obtained more than 6200 original infrared and visible images from the videos collected.

The experiments were performed using an Intel(R) Core (TM) i7-7700HQ CPU@2.80GHz (California, USA), NVIDIA GeForce GTX1050 (California, USA), and Windows8 64-bit operating system (Washington, DC, USA). The processing software sets include MATLAB2018a (MathWorks, USA), Visual Studio 2017 (Microsoft, USA), and OpenCV3.4.1 library (Intel, USA).

### 5.1. Evaluation of Frame Selection

The proposed FSIFD and the retained frames extraction based on clustering (RFEC) [9] are compared experimentally in terms of video continuity, frame numbers, and playing duration.

Considering the Fast video1 and Slow video1 in suburban road scene as an example, the objective indicators of videos before and after frame selection are shown in Table 1.

It is evident from Table 1 that the NCIEs of the retained frame sequence are smaller than that of the original video at different driving speeds. This indicates that the overall correlation degree of the content is reduced after frame removal. The DFR is higher in the slow-speed vehicle video, because the difference between the frames is small and the redundant frames are more numerous.

In slow video1, the proposed FSIFD discards 293 frames at the cost of a 0.2% reduction in NCIE, and the DFR is as high as 79%. The frames to be fused only account for 21% of the original video. The RFEC discards 307 frames at the cost of NCIE by 0.3%, and the DFR is as high as 83%.

In fast video1, the NCIE only reduces by 0.3% when 144 frames are discarded in the proposed method. In addition, the DFR is 38%, and the frames to be fused is 62% of the original video. In the RFEC, 287 frames are discarded, and the DFR is as high as 77%, but NCIE is reduced by 2.9%. This is because the RFEC does not consider the motion between adjacent video frames, resulting in discontinuity and flickering of the selected frames.

The above analysis shows that the proposed FSIFD ensures a high DFR on the premise of continuous video content at different vehicle speeds.

It is also evident from Table 1 that under the same frame rate, the playing durations of the retained frame sequence are much shorter than that of the original videos. The difference in playing durations is larger due to the higher DFR of the slow video, indicating that the frame selection can improve the efficiency, but will change the video duration. Therefore, it is necessary to insert frames after selecting frame fusion to restore the playing durations.

### 5.2. Evaluation of Frame Fusion

The original visible and infrared images of a small halation area scene on an urban main road and a large halation area scene on suburban road are fused by the improved ©-Curvelet algorithm. The fusion results are shown in Figure 6 and Figure 7, respectively.

As presented in Figure 6, in a low-illumination environment at night, the high brightness of the halation area causes the illumination of the remaining dark areas to be further reduced in the visible image, and information such as pedestrians, buildings and road contours are more difficult to observe. There is no halation in the infrared image; the contours of pedestrians and buildings can be seen, while the color information is missing and the contrast is low. The fused image eliminates halation, retains color, and enriches details such as the lanes and road edges, which meets the characteristics of human eyes sensitive to color.

As presented in Figure 7, there is almost no other road information except for halation of the oncoming headlights in the visible image. The infrared image is not affected by halation, and the road environment is visible, but the resolution is low and the color is missing. The fused image has higher halation elimination and richer details, which are more conducive to human observation.

In summary, the halation of visible images seriously affects driving safety. The infrared image has poor visual effect when used to assist driving at night. The fused image is more suitable for human visual observation in the night halation scene.

### 5.3. Evaluation of Frame Interpolation

The proposed TMBAMC and the motion vector estimated by LSTMTA [11], OFFRN [12] and MVEBMS [15] are compared in terms of the accuracy of the number of frame interpolation and the synchronization of the content.

#### 5.3.1. Accuracy of Number of Frame Interpolation

The retained frame sequence of a slow video in a suburban road scene is considered as an example. The result of frame interpolation is shown in Figure 8. The statistics of the number of frame interpolations are shown in Table 2.

As presented in Figure 8, Table 1 and Table 2, the total frames reach 377, 311, and 386 after frame interpolation by LSTMTA, OFFRN, and MVEBMS, respectively. The frames of LSTMTA and MVEBMS are 6 and 15 frames more numerous than those available in the original video, respectively, and the playing duration is 0.24 s and 0.6 s longer; in contrast OFFRN is 60 frames less numerous, and the playing duration is 2.4 s shorter.

Among the 77 groups of interpolation results by LSTMTA, 57 groups are different from the original video, and the error rate is as high as 74%. In 32 groups, the interpolated frames are more numerous than in the original sequence, and the content is almost no different, resulting in a video stalling phenomenon. In another 25 groups, the interpolated frames are less numerous than in the original frames, which leads to a shorter playing duration and a video flicker phenomenon.

OFFRN and MVEBMS have 46 and 59 groups of interpolation results that are inconsistent with the original video, respectively, among which 12 and 38 groups are more numerous than the original video, 34 and 21 groups are less numerous, and the error rate reaches 60% and 77%.

This is because MVEBMS does not consider the influence of content differences between the adjacent reserved frames, and always inserts the fixed number of frames with the same content difference between two frames. OFFRN is based on the assumption that the object has a small displacement, and it has an ability to sense the motion changes between adjacent frames in slow video, so the number of inserted frames is more accurate than LSTMTA and MVEBMS.

The proposed TMBAMC reconstructed frames according to time stamp, number of frames and playing duration of the interpolated video are consistent with that of the original video. Moreover, the interpolated frames have the same visual effect as the original sequence.

#### 5.3.2. Synchronization of Frame Interpolation Content

When the number of frames inserted by four algorithms are consistent with that of the original video, the quality of the interpolated frame and the video continuity are further evaluated for the slow video in urban main road scene and the fast video in suburban scene.

Figure 9 shows two adjacent retained frames of the slow video in the urban main road scene, i.e., the original frame between frames 34 and 38, as well as the intermediate frame restored by LSTMTA, OFFRN, MVEBMS and the proposed TMBAMC.

As presented in Figure 9, the number of building floors in the first frame interpolated by LSTMTA is more than that of original frame 35, and the position of the car is ahead of frame 35. As compared with the original frame 36, the height of the trees is lower in the second interpolated frame. In addition, the position of the car and the amplitude of the pedestrian’s arm swing are ahead of frame 36. However, in the third interpolated frame, the position of car and the amplitude of the pedestrian’s arm swing lag behind frame 37.

The position of the car in the first frame interpolated by the OFFRN is slightly ahead of the original frame 35. In comparison with frame 36, the position of the car and the amplitude of the pedestrian’s arm swing are both ahead in the second interpolated frame.

The height of the trees in the first frame interpolated by the MVEBMS is lower than that of original frame 35, while the number of building floors is greater. In addition, the position of the car and the amplitude of the pedestrian’s arm swing lag behind frame 35. In comparison with frame 36, the position of the car and the amplitude of the pedestrian’s arm swing are both ahead in the second interpolated frame.

This shows that the content of the interpolated frame is different from that of the original frame in LSTMTA, OFFRN and MVEBMS, and the effect is poor. The interpolated frames constructed by the proposed TMBAMC are almost similar to the original video in subjective vision, and the interpolated frames have higher quality.

Figure 10 shows two adjacent retained frames of the fast video in suburban road scene, i.e., the original frame between frame 44 and frame 49, as well as the intermediate frame restored by LSTMTA, OFFRN, MVEBMS and the proposed TMBAMC.

As presented in Figure 10, the position of the first car or the relative positions of the two cars in front in the frames interpolated by LSTMTA, OFFRN and MVEBMS is different from that of the corresponding frames in original video to varying degrees. However, the interpolated frame constructed by the proposed TMBAMC is almost similar to the original frame.

In conclusion, at different vehicle speeds in various scenes, the quality of interpolated frames constructed by the proposed TMBAMC is higher as compared to LSTMTA, OFFRN and MVEBMS in terms of subjective evaluation.

In order to evaluate the quality of the interpolated frame and the continuity of the interpolated video objectively, the content synchronization is reflected by the vector angle *α* between the interpolated and the corresponding original frame. Ideally, when *α* = 0, the two frames are completely synchronized. In practice, it is considered that when *α* < 1, the content is synchronized; when *α* ≥ 1, the content is not synchronized; and when *α* > *τ*_op_, it is discontinuous.

The difference between the interpolated and the original frame of the slow video in urban main road scene is shown in Figure 11.

It is evident from Figure 11 that there are 22 frames with *α* greater than 1 in the first 50 frames processed by LSTMTA, and the asynchronization rate between the interpolated frame and the corresponding original frame is as high as 44%. Especially, *α* reaches 1.82, 1.88, 1.87 and 1.89 in frame 21, 36, 49 and 50 respectively, and is greater than *τ*_op_ set by frame selection, which indicates that the video content is interrupted after frame insertion. The reason is that the quality of the subsequent interpolated frames is affected by the constructed interpolated frames in LSTMTA, which can cause error accumulation.

The overall fluctuation of *α* in OFFRN is less than that in LSTMTA, and all values of *α* are less than *τ*_op_, indicating that the content of interpolated frames is continuous. There are 7 frames in which *α* is greater than 1, and the asynchronization rate is 14%, which is 30% lower than that in LSTMTA. It indicates that OFFRN is superior to LSTMTA in the quality of frame interpolation for the slow video, meeting the assumption of small motion.

In MVEBMS, *α* reaches 1.88 and 1.81 in frame 14 and 35, respectively, and the video content is interrupted. *α* fluctuates greatly from frame 13 to 16, as well as from frame 35 to 36, is greater than 1, and the asynchronization rate is 12%, which is reduced by 32% and 2% compared with LSTMTA and OFFRN, respectively. The reason is that MVEBMS considers the motion between two adjacent reserved frames, so the effect of frame interpolation is more reliable.

However, *α* fluctuates slightly in the whole sequence and is always less than 1 for the proposed TMBAMC, indicating that the content is synchronous and continuous, and the effect is more stable. The mean of *α* is 0.96, 0.68 and 0.86 in LSTMTA, OFFRN, MVEBMS respectively, while it is 0.54 in the proposed TMBAMC, indicating that the frames reconstructed by the proposed algorithm have higher quality. The reason is that the proposed TMBAMC estimates the motion vector based on the time mark and the content difference between the original frames, and have a higher expression for original video content, so the constructed frame is more similar to the original frame.

The difference between the interpolated frame and the original frame of the fast video in the suburban road scene is shown in Figure 12.

It is evident from Figure 12 that *α* is always less than 1, and shows small fluctuation in the whole sequence for the proposed TMBAMC, indicating that the video content is synchronous and continuous after frame interpolation.

However, *α* fluctuates greatly for LSTMTA, OFFRN and MVEBMS. The frames with *α* greater than 1 account for 46%, 54% and 30% of the total sequence, respectively, and the frames with *α* greater than *τ*_op_ account for 4%, 10% and 10%. Compared with the slow video in the urban main road scene, the discontinuous of video content and the stalling phenomenon are more serious. As a result, the effect of frame interpolation has further deteriorated.

The means of *α* in the LSTMTA, OFFRN and MVEBMS are 1.11, 1.18 and 0.97, respectively, while it is 0.57 in the proposed TMBAMC. Compared with the slow video in urban main road scene, the mean of *α* increases by 15.63%, 73.53%, 12.79% and 5.56%, respectively. Among them, due to the limitation of OFFRN based on ideal assumption, the frame insertion effect for the fast video is significantly different from that for the slow video, and its universality is low.

From the above analysis, it can be obtained that the faster the vehicle speed, the lower the quality of the interpolated frames. Under the same conditions, the change in the mean of *α* is significantly smaller in the proposed TMBAMC, indicating that the algorithm has better quality of frame interpolation and stronger adaptability.

#### 5.3.3. Evaluation of Anti-Halation Performance of Video Fusion

Considering the fast and slow videos in the suburban road and urban main road scenes as examples, the proposed method and the frame-by-frame fusion method are used to experiment the video fusion efficiency. The frame rate (FPS), time complexity T(*n*) and space complexity S(*n*) of fusion video are shown in Table 3.

It is evident from Table 3 that the average FPS of frame-by-frame fusion video is 1.03 in four videos, and that of the proposed method is 6.11, which is about six times higher, indicating that the proposed method can effectively improve the efficiency of video fusion. Under the same S(*n*), T(*n*) of the proposed method is reduced by one magnitude compared with frame-by-frame fusion, indicating the proposed method effectively reduce the computational complexity.

As the driving speed is the same, there is little difference in the frame rate of fused video for various scenes, indicating that the fusion efficiency is less affected by the scenes. In the same scene, the fused video with faster speed has a lower frame rate. This is because the DFR of fast video is lower during frame selection, and there are more frames to be fused.

## 6. Conclusions

The anti-halation method of video fusion proposed in this work effectively solves the lag caused by the frame-by-frame fusion of infrared and visible images. The designed frame selection fusion strategy discards the redundant frames to the greatest extent on the premise of continuous video content, reduces the number of frames to be fused, and improves the processing efficiency of video fusion. The proposed TMBAMC ensures that the video content is continuous and synchronized after frame insertion, and the duration is equal to that of the original video, which solves the phenomena of video stalling and flickering in the MVEBMS. The anti-halation method based on infrared and visible video fusion proposed in this work is applied to a night halation scene, which has a good halation elimination effect and helps to improve the efficiency of video fusion.

## Figures and Tables

**Figure 1 sensors-22-07494-f001:**
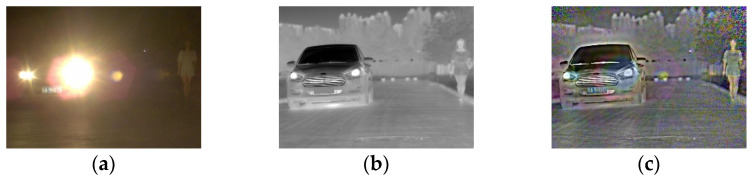
The original image and the corresponding fused image. (**a**) Visible image; (**b**) infrared image; (**c**) fused image.

**Figure 2 sensors-22-07494-f002:**
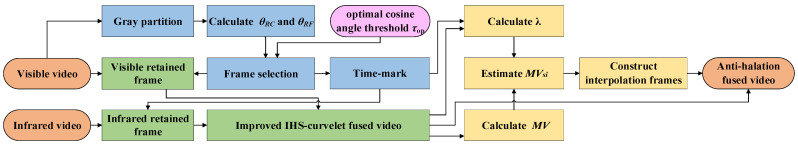
The overall block diagram of the proposed method.

**Figure 3 sensors-22-07494-f003:**
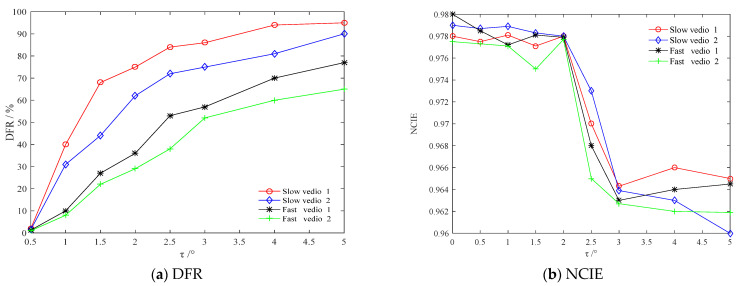
The relation of DFR and NCIE with *τ*.

**Figure 4 sensors-22-07494-f004:**
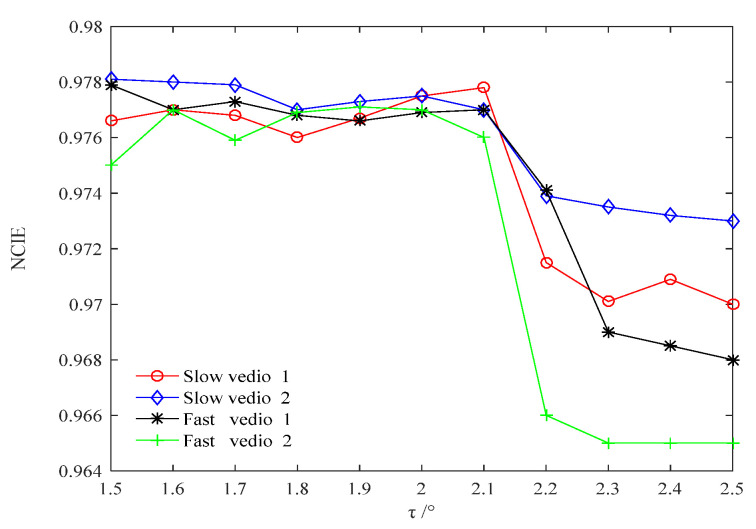
The NCIE in *τ* ∈ [1.5,2.5].

**Figure 5 sensors-22-07494-f005:**
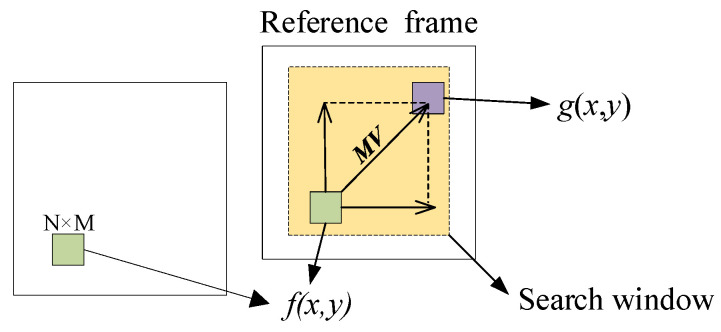
A schematic diagram of block matching search.

**Figure 6 sensors-22-07494-f006:**
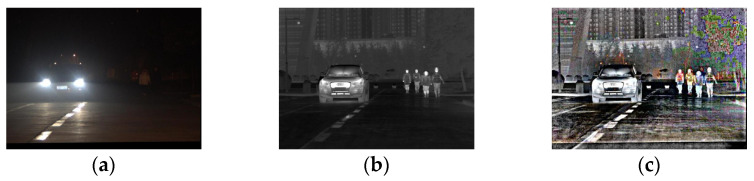
The original and fused images of small area halation scene on urban main road. (**a**) Visible image; (**b**) infrared image; (**c**) fused image.

**Figure 7 sensors-22-07494-f007:**
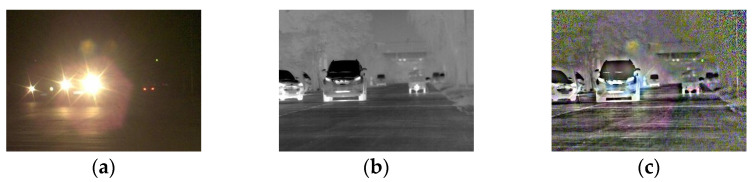
The original and fused images of the large halation area scene on suburban road. (**a**) Visible image; (**b**) infrared image; (**c**) fused image.

**Figure 8 sensors-22-07494-f008:**
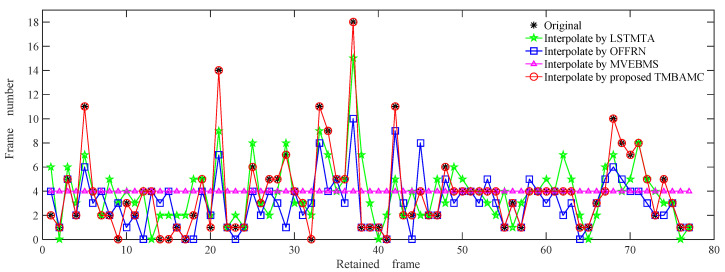
The result of frame interpolation of slow video in suburban scene.

**Figure 9 sensors-22-07494-f009:**
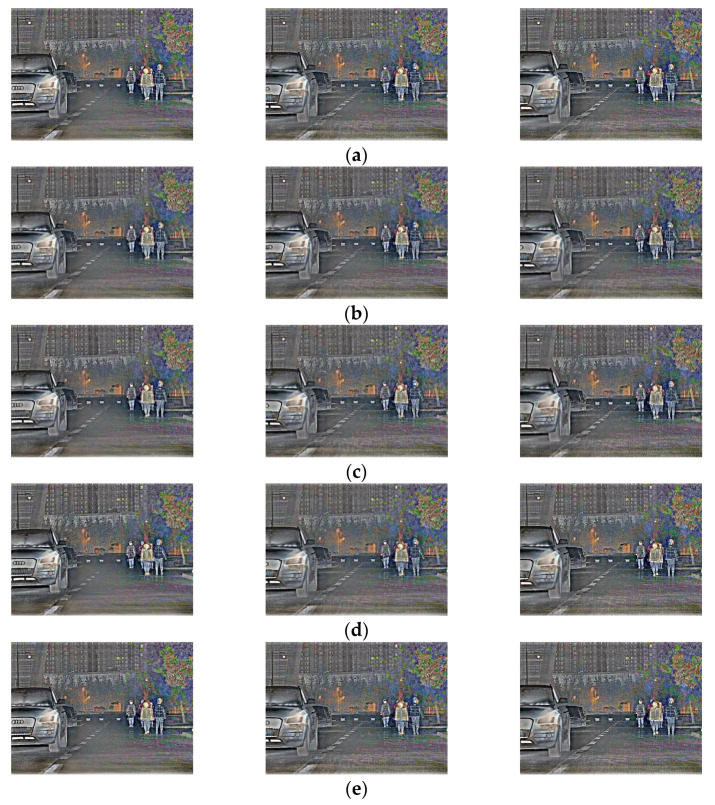
The original and the interpolated frames between frames 34 and 38 of the slow video in urban main road scene. (**a**) Original frames; (**b**) interpolated frames by the LSTMTA; (**c**) interpolated frames by the OFFRN; (**d**) interpolated frames by the MVEBMS; (**e**) interpolated frames by the proposed TMBAMC.

**Figure 10 sensors-22-07494-f010:**
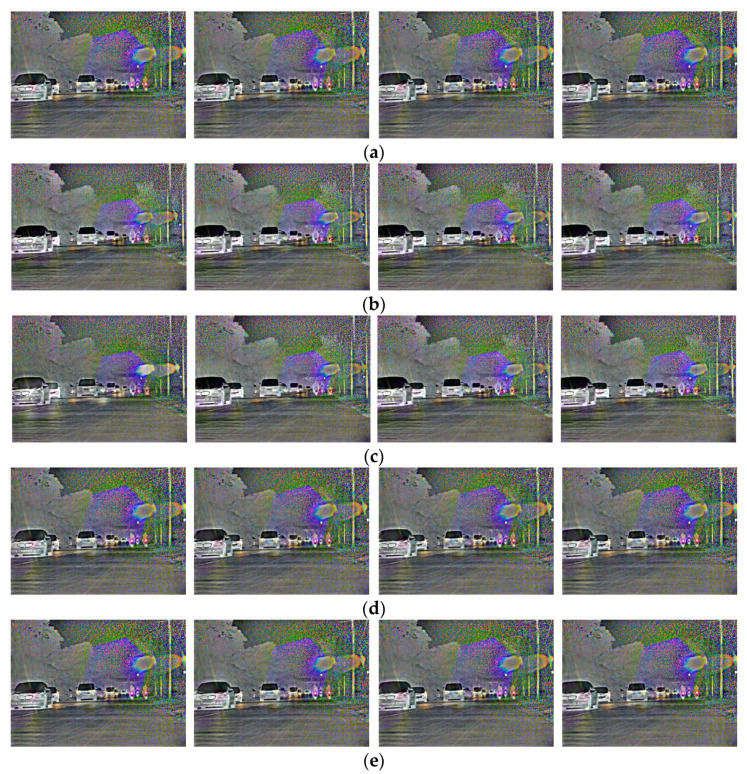
The original and the interpolated frames between frames 44 and 49 of the fast video in suburban road scene. (**a**) Original frames; (**b**) interpolated frames by the LSTMTA; (**c**) interpolated frames by the OFFRN; (**d**) interpolated frames by the MVEBMS; (**e**) interpolated frames by the proposed TMBAMC.

**Figure 11 sensors-22-07494-f011:**
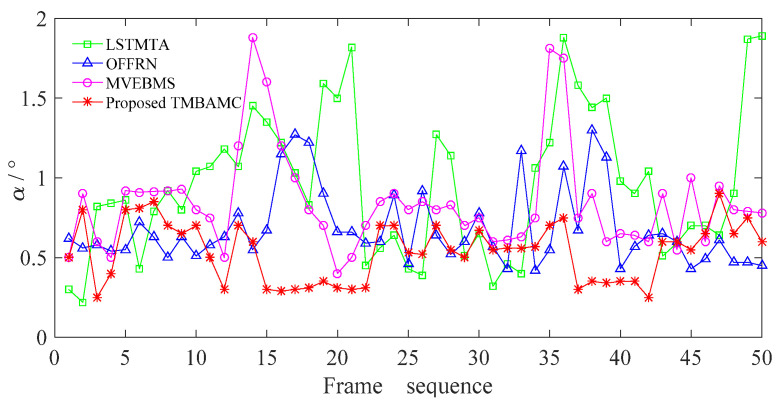
The variation of *α* for slow video in urban main road scene.

**Figure 12 sensors-22-07494-f012:**
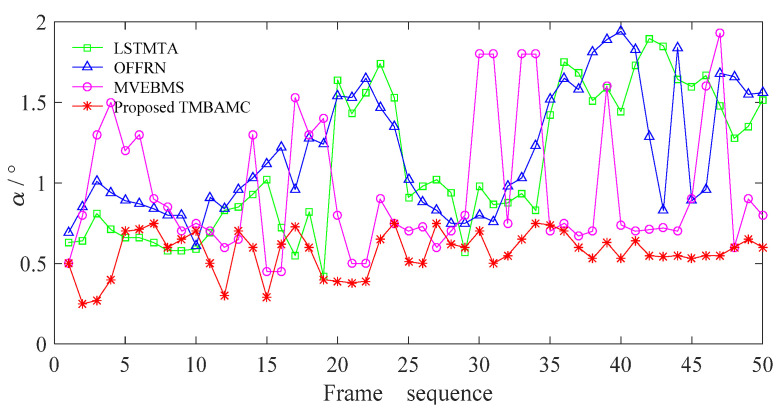
The variation of *α* for fast video in suburban road scene.

**Table 1 sensors-22-07494-t001:** The objective indicators of suburban road video sequences.

Video	Original Sequence			Retained Sequence by FSIFD			Retained Sequence by RFEC		
	NCIE	Frames	Duration	NCIE	Frames	Duration	NCIE	Frames	Duration
Fast video1	0.978	371	14.84 s	0.976	78	3.12 s	0.975	64	2.56 s
Slow video1	0.980	375	15.00 s	0.977	231	9.24 s	0.952	88	3.52 s

**Table 2 sensors-22-07494-t002:** The statistics of the number of frame interpolation.

Algorithm	N_IF_ * = N_OF_		N_IF_ ≠ N_OF_			Total Frames	Duration
	Groups	Frames	Groups (N_IF_ > N_OF_)	Groups (N_IF_ < N_OF_)	Frames		
LSTMTA	20	69	32	25	230	377	15.08 s
OFFRN	31	67	12	34	166	311	12.44 s
MVEBMS	18	72	38	21	236	386	15.44 s
TMBAMC	77	293	0	0	0	371	14.84 s

* N_IF_ represents the number of interpolation frames; N_OF_ represents the number of original frames.

**Table 3 sensors-22-07494-t003:** The FPS, T(*n*) and S(*n*) of different fusion videos.

Experiment	The Frame-By-Frame Fusion			The Proposed Method		
	FPS	T(*n*)	S(*n*)	FPS	T(*n*)	S(*n*)
slow video on suburban road	1.20	O(n^3^)	O(n)	6.83	O(n^2^)	O(n)
fast video on suburban road	0.90			5.88		
slow video on urban main road	1.08			6.65		
fast video on urban main road	0.95			5.72		

## Data Availability

The datasets generated during this study are available from the corresponding author on reasonable request.

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
