# Peer review of "Night Vision Anti-Halation Method Based on Infrared and Visible Video Fusion"

_sensors, 2022, doi:10.3390/s22197494_

Round 1
Reviewer 1 Report
In order to address the discontinuity caused by the direct application of the infrared and visible image fusion anti-halation method to a video, an efficient night vision anti-halation method based on video fusion is proposed. The designed frame selection based on inter-frame difference determines the optimal cosine angle threshold by analyzing the relations of cosine angle threshold with nonlinear correlation information entropy and de-frame rate. The proposed time-mark-based adaptive motion compensation constructs the same number of interpolation frames as the redundant frames by taking the retained frame number as a time stamp. At the same time, considering the motion vector of two adjacent retained frames as the benchmark, the adaptive weights are constructed according to the interframe differences between the interpolated frame and the last retained frame, then the motion vector of the interpolated frame is estimated. The experimental results show that the proposed frame selection strategy ensures the maximum safe frame removal under the premise of continuous video content at different vehicle speeds in various halation scenes. The frame numbers and playing duration of the fused video are consistent with that of the original video, and the content of the interpolated frame is highly synchronized with that of the corresponding original frames. The average FPS of video fusion in this work is about 6 times that in the frame- by-frame fusion, which effectively improves the anti-halation processing efficiency of video fusion. Generally, this is a good work. It can be accepted if the authors can consider the following issues: 1. The motivations and original contributions should be well organized in the introduction part. 2. How did the get the dataset of video? 3. For Fig 9, please explain why the proposed method can lead to better results? 4. More related works are welcome to enrich the literature review such as Transnational image object detection datasets from nighttime driving, Large image datasets: A pyrrhic win for computer vision? BIM, machine learning and computer vision techniques in underground construction: Current status and future perspectives 5. How about the computational load of the proposed method?
Reviewer 2 Report
The motivation of the study is not well driven by the facts. The novelty of the work is also marginal. There is a few analysis on the theoretical stability of the fused data. A proper comparative analysis is missing. The authors fail to substantiate their work by comparing it with the state of the art works. Also, there is no analysis about the fused and un-fused results. how does the fusion really outperforms the unfused ones. The statistical quantification is insufficient to arrive at a conclusion about the model.
Round 2
Reviewer 2 Report
The paper may be proof read once again to perform a thorough spell and grammar check.